# Species specific marker genes for systemic defence and stress responses to leaf wounding and flagellin stimuli in hybrid aspen and silver birch

**Kārlis Blūms**[1], **Baiba Krivmane**[2], **Maryna Ramanenka**[2], **Roberts Matisons**[2], **Dainis Ruņģis**[2], **Mārtiņš Zeps**[2], **Zigmunds Orlovskis**[1]*

1 Latvian Biomedical Research and Study Centre, Rīga, Latvia, 2 Latvian State Forest Research Institute "Silava", Salaspils, Latvia

* zigmunds.orlovskis@biomed.lu.lv

## Abstract

In Northern Europe, climate warming is driving the northward expansion of deciduous tree species such as aspen and silver birch, while simultaneously intensifying biotic stress from pests and pathogens. This creates an urgent need for improved understanding of molecular defence mechanisms underlying stress resistance and resilience in temperate forest trees, as a basis for the development of innovative biotechnological approaches. However, progress in this area remains limited by the lack of reproducible experimental systems and well-characterized molecular markers for systemic defence responses in deciduous tree species. In this study, we aimed to identify and validate known plant defence gene markers associated with systemic stress responses in hybrid aspen and silver birch to support future functional research. Using sequence mining and phylogenetic analyses, we identified homologues of biotic stress-response genes in the genomes of both species. We then employed *in vitro* propagated tree clones to assess defence gene activation in distal leaves following systemic signal induction by leaf wounding and bacterial flagellin treatment at 4 and 24 hours post-induction. We identified *LOX2, MPK3*, and *EIN2* as early wounding-responsive genes in silver birch, while *JAZ10* together with *EIN2* showed robust induction in hybrid aspen in response to the combined effects of wounding and flagellin. Collectively, these findings establish a reproducible *in vitro* framework for validating stress responsive genes and provide a foundation for future studies of systemic signalling, tree–microbe interactions, and stress resilience in ecologically and economically important forest tree species.

## 1. Introduction

Accelerating climate change, intensifying weather extremes and their legacy effects, such as outbreaks of pests and diseases [1,2], are subjecting trees to increasingly

Data availability statement: All relevant data are within the manuscript and its Supporting information files.

Funding: This work was supported by Latvian Council of Science Fundamental and Applied Research grant lzp-2022/1-0283. The funders had no role in study design, data collection and analysis, decision to publish, or preparation of the manuscript.

Competing interests: The authors have declared that no competing interests exist.

intense stresses [3]. These stresses can exceed the adaptive capacity (i.e., phenotypic plasticity) of local tree populations [4], emphasizing the urgent necessity for adaptive management strategies [5] to cope with growing environmental challenges [6]. Given the long life cycle of trees and the stresses associated with large-scale environmental changes [7,3], effective adaptive measures should be self-sustaining and grounded in a thorough understanding of ecological, ecophysiological, and biochemical processes [8]. Examples of such measures include locally finetuned management [9,10], as well as the inoculation of plants with beneficial microbes, such as mycorrhizal fungi, which have been suggested as promising strategies to enhance forest resilience against increasing environmental stresses [11,12,13].

In the eastern Baltic region, climate-driven shifts in forest species distribution are anticipated, with deciduous species expected to push conifers northward [14]. Under these conditions, the prevalence of silver birch (*Betula pendula*) and aspen (*Populus tremula*) — both currently of high commercial importance — is projected to increase [15,16]. However, both species are sensitive to climate change [17,18], and thus, an intensification of biotic legacy effects, such as insect pest outbreaks following climatic disturbances, is also expected [19,3]. Additionally, the use of specific, highly productive genotypes — such as the most economically valuable birch clones or superior hybrid aspens — is recommended for commercial forestry [20,16]. This underscores the need to identify innovative biotechnological approaches for improving stress tolerance in selected high yielding genotypes and to enhance understanding of tree stress responses at a molecular level [21,22,23].

Molecular stress responses have been extensively studied in herbaceous crop plants and model species such as *Arabidopsis thaliana*, however, trees with their large and complex genomes have been comparatively understudied and often limited to genus *Populus* [24]. This has hitherto limited functional studies on the genetic mechanisms involved in biotic stress responses in other economically important tree species. Nevertheless, recent advances in sequencing the genomes of important forestry species such as silver birch [25] or several aspen species [26] have catalysed recent progress in elucidating the molecular basis of tree responses to abiotic stresses [21]. Similar to *Arabidopsis,* tree responses to chilling (frost), heat and drought, are likely mediated by abscisic acid (ABA), ROS and $Ca^{2+}$ signalling, mitogen-activating protein kinases (MAPK), heat-shock proteins (HSP), MYB and bZIP family transcription factors (among others), resulting in antioxidant production, changes in membrane fluidity and repair (reviewed by [21,24]). Still, reliable stress responsive genes for studying tree responses to herbivore attack or microbial infections remain to be characterised for most tree species.

This study aimed to identify plant defence gene homologues associated with systemic stress responses in hybrid aspen and silver birch and to validate their activation using *in vitro* propagated tree clones subjected to controlled biotic stress cues. Leaf wounding and application of the immunogenic bacterial peptide flagellin were used to mimic mechanical damage caused by chewing herbivores and bacterial elicitor perception via the pattern-triggered immunity receptor FLS2 [27], respectively. To this end, we mined the hybrid aspen and silver birch genomes for genes involved in

jasmonic acid–mediated wound responses, including *LOX* and *JAZ* family members [28], as well as for genes associated with salicylic acid–related defence responses to bacterial elicitors, including *FRK1, NHL10, MPK3, PAD4*, and *MYB51* [29,30]. Using *in vitro* propagated tree clones as a reproducible experimental platform, we characterized the activation of selected defence genes in distal leaves following induction of systemic signals by local leaf wounding and flagellin treatment at defined time points. Here, we present an *in vitro*-based framework for the initial validation of stress responsive genes of systemic defence signalling in birch and aspen, providing a foundation for future studies of intra- and inter-tree communication and biotic stress responses.

## 2. Materials and methods

### 2.1 In-vitro propagation of tree clones

Silver birch and hybrid aspen clone cultures were obtained from the clonal collection of the LSFRI "Silava" plant physiology laboratory. For hybrid aspen *Populus tremuloides × tremula*, clone "44" was selected, as it shows superior field performance [31]. For silver birch, clone "54-146-143", which shows above average field performance and exceeds wild populations by 7–30% in volume growth [8,32], was used. The clones were cultivated *in vitro* on 1X Murashige and Skoog (MS) media, supplemented with MS vitamins, MS micronutrients, 20 g/L sucrose, 6 g/L of agar and 0.1 mg/L indole-3-butyric acid (IBA) at pH = 5.8 [33].

Each clone (plantlet) was cultivated individually in a 150 mL jar filled with 15 mL of the media. During cultivation, the jars were covered with an aluminium foil cap to prevent microbial contamination whilst enabling air flow and easy access to the plantlet for manipulation and stress treatments. To ensure controlled environmental conditions, all the clone plantlets were grown in a climate chamber maintained at 30−40% relative humidity and 25 °C on four multi-layer shelf systems equipped with luminaries. All clones were grown under the same illumination with a photon flux density of 110 ± 10 µmol m$^{-2}$ s$^{-1}$, for the wavelengths ranging from 400 to 750 nm with 16 h light and 8 h dark photoperiod to imitate long-day conditions of the growing period. For each tree species, a total of 32 plantlets were cultivated for the assessment of stress responses in an orthogonal time-course experimental design, allowing four biological replicates per treatment and observation combination.

### 2.2 Stimulation of local leaves and sample collection

To assess defence-related gene expression, the plantlets were subjected to three different treatments: a chemical stimulus with 22 amino acid fragments of bacterial flagellin (*flg22*), mechanical injury to leaves and the combination of both. These stimuli were chosen to imitate herbivore attack (mechanical injury) or the attack of a bacterial pathogen (*flg22*) to activate defence responses related to jasmonic and salicylic acid pathways, respectively [28,34,35]. The mechanical injury was applied by squeezing the leaf with forceps in three parallel lines, while the flagellin stimulus was applied by pipetting 5 µL of 1 µM *flg22* solution (in water), on the abaxial surface of the leaf. For the combined treatment, *flg22* was applied on the wounded abaxial surface of the leaf. As the control, 5 µL of distilled nuclease-free water was used.

To minimize potential diurnal variation in gene expression, all stimuli were applied at noon (12:00), 4 h after the onset of grow chamber lighting. The stimuli, including the control, were applied to the third fully unfolded leaf from the apex, marking the stimulated leaf on the outside of the jar. Plantlets were treated in a sterile environment of laminar flow before returning to the climate chamber. For the gene expression analysis, a non-stimulated (naïve) leaf – second fully unfolded leaf from the apex – distal from the treated leaf was collected 4 and 24 h after the treatment to characterise the short-term dynamics of systemic plant responses. To preserve RNA, the samples were flash-frozen in liquid nitrogen and stored at −80 °C.

### 2.3 RNA extraction and cDNA synthesis

Total RNA was extracted following a modified CTAB extraction protocol from Rubio-Piña and Zapata-Pérez [36] adapted for use with woody plant tissue (detailed protocol in S1 File). The total RNA concentration and purity (260/280 and 260/230 ratios) were determined with the NanoDrop2000 spectrophotometer (Thermo Scientific). 500 ng of total RNA

(260/280 and 260/230 ratios between 1.8 and 2.2) were processed for reverse transcription reaction using the Thermo Scientific Maxima First Strand cDNA synthesis kit according to manufacturer instructions. cDNA synthesis reaction was carried out using a BIO-RAD T100™ thermal cycler and the reaction product was subsequently diluted to 800 μL in nuclease-free $H_2O$.

### 2.4 Primer design

In order to observe potential differences in defence related gene expression in naïve systemic plant tissue (distal leaf), 8 different genes with a well described role in *Arabidopsis thaliana* defence mechanisms were chosen as the qPCR targets (S1 Table); *EIN2, FRK1, PAD4, LOX2, MPK3, NHL10, JAZ10, MYB51*, with *ACT2* chosen as a constitutively expressed reference gene for both taxa. Using gene specific protein sequences obtained from The Arabidopsis Information Resource (TAIR) [37], BLAST was used to find potential defence gene homologues in the genomes of both tree taxa. To increase the chances of finding optimal gene homologues, BLAST search was performed in two databases for each taxon: the National Centre for Biotechnology Information (NCBI) database was used for both taxa (*Populus tremula x tremuloides taxid:47664; Betula pendula taxid:3505*), while Comparative Genomics (CoGe) [38] and PlantGenIE [39] were used for birch (*Betula pendula scaffold assembly id35079, id35080*) and hybrid aspen (*Populus tremula x tremuloides v10*), respectively. The BLAST search results with the highest homology identifiers (query coverage, percent identity and e-value) were chosen as the potential homologues of each gene and subjected to reciprocal BLAST against *A. thaliana* and other taxa included in the NCBI database.

A complementary phylogenetic analysis of the selected tree homologues was performed using the MEGA (v11.0.13) Maximum likelihood algorithm (bootstrap = 500). In addition to the potential tree homologues, additional *A. thaliana*, *Medicago truncatula* and *Marchantia polymorpha* sequences from NCBI and MarpolBase were included in the phylogenetic trees. Based on the *in silico* gathered data, a single potential homologue was selected for each gene as displayed in S1 and S2 Figs. Using NCBI Primer–BLAST, a primer pair was designed for each selected gene sequence. All primer pairs were designed with $T_m$ of around 60 °C (with ±1 °C as the maximum deviation) and PCR product size of 100–300 nt, as reported in S1 Table.

### 2.5  rt-qPCR analysis

The qPCR reactions were carried out using the Thermo Scientific Maxima SYBR Green/ROX qPCR Master Mix (2X) kit, with each reaction containing 2 μL of 0.3 μM primer mix, 5.5 μL of sample cDNA solution and 7.5 μL of the Maxima SYBR Green/ROX qPCR Master Mix (2X). rt-qPCR was performed on *ViiA™ 7 Real-Time PCR System (Applied Biosystems)* using 3-step cycle (initial denaturation at 95°C for 10 min; denaturation at 95°C for 15s; annealing at 60°C for 30s; extension at 72°C for 30s) and analysed with the native *QuantStudio 7 Pro 1.6.1* software. For each sample, three technical replicate qPCR reactions were carried out with each primer pair. The resulting average Ct value was then used in further calculations. Amplicons were checked for single peaks in their melting curves (S3 Fig) as well as visualized on an electrophoresis gel for the presence of a single band at the expected amplicon length (S4 Fig).

Primer efficiency (E%) was calculated using the formula $E(\%) = \left( \frac{-1}{10^{Slope}} - 1 \right) \times 100$, with slope being obtained by plotting the $log_{10}$ values of 10-fold serial dilutions from 1ng to 100ag of purified cDNA (S5 Fig). The expression of each test gene was normalized by the expression of a single reference gene (*ACT*) using the formula $\Delta Ct = E_{ref}{}^{\wedge}Ct_{ref} / E_{test}{}^{\wedge}Ct_{test}$ where $E_{ref}$ – primer efficiency of the reference gene, $E_{test}$ – primer efficiency of the test gene, taken to the power (^) of $Ct_{ref}$ – Ct value of the reference gene, and $Ct_{test}$ – Ct value of the test gene, respectively. In total, four expression values were obtained for each gene, for all stimuli groups and at each time-point for both birch and hybrid aspen. To determine change in defence gene expression relative to the water control, log2 fold change was calculated using the formula $\Delta\Delta Ct = log2(x/y)$, where x – the average normalized expression (ΔCt) of a select treatment and y – the average normalized expression (ΔCt) of the control group. Analysis was done separately for each timepoint.

## 2.6 Leaf wetting experiments

Leaf wetting experiment was performed according to the method by Limm et al. [40]. Single leaves were cut from the plant and petioles sealed with hydrophobic Parafilm to prevent water loss or entry through the petiole tip. Leaves were fully submerged into dH$_2$O, 0,2% (w/v) rhodamine B solution or the rhodamine B solution with 0,001% (v/v) Tween80. Foliar water uptake (FWU) into apoplast was calculated by weighing the mass of the detached leaf (with parafilm) and calculating using the formula FWU = (M2 − M1) − (M4 − M3), where M1- mass of leaf before submergence, M2- mass of leaf submerged for 3h then blotted on tissue to remove excess water, M3 - mass of leaf after M2 measurement and air-drying for 5 min, M4 – mass of leaf after the M3 measurement resubmerged for 1s, then immediately blotted on tissue to remove excess water. Leaf surface contact angle with a 5 µL drop of 1µM *flg22* was determined by taking photographs with 100 mm 1X magnification macro lens at the minimal focal distance (30 cm) at F2.8 and 1/60s, then analysed with the ImageJ plugin Contact Angle.jar.

## 2.7 Promoter sequence analysis

5k upstream region from TTS for *AtLOX2* (AT3G45140), *AtEIN2* (AT5G03280) were obtained from the TAIR database, for *BpLOX2* (Bpev01.c0523.g0011), *BpEIN2* (Bpev01.c0990.g0003) form CoGe database and *PttLOX2* (Potrx066157g26369.5), *PttEIN2* (Potrx046169g13695.4), *PpLOX2* (Potri.001G015300.1) from PlantGenIE database. TCP4 (MA1035.1; TFmatrixID_0423), TCP20 (MA1065.1; TFmatrixID_0424), TCX5 (UN011.1) binding motifs were obtained from JASPAR [41] and PlantPan [42] databases. Promoter alignment and identification of transcription factor binding sites was performed with PlantPAN v4.0 built-in Promoter analysis tool and Cross species Blast2SEQ tool.

## 2.8 Statistical analysis

To assess differences in gene expression between treatment groups, a two-way, single factor analysis of variance (ANOVA) was performed along with Tukey HSD post-hoc tests in cases of significant differences, using base R. In cases where ANOVA assumptions were violated, non-parametric Kruskal-Wallis tests and subsequent post-hoc Dunn tests were performed. Kruskal-Wallis tests were done using base R, while the "dunn.test" package ([43], R package version 1.3.6. ) was used for the Dunn post-hoc tests. A significance level α of 0.05 was used for all tests. Heatmaps were generated using the *pheatmap* function from the "pheatmap" package ([44], R package version 1.0.12) in R with centroid linkage clustering based on Euclidean distance. To assess the interaction between wounding or *flg22* treatments and the different time-points of gene-expression across the gene panel, a multivariate linear model was used where gene Fold Change ~ Treatment * GeneID * Time-point. The statistical models were checked for compliance with assumptions via diagnostic plots. Data analysis was conducted in R version 4.5.0. (R Core Team). The full R-script used in statistical analysis and heatmap generation is included in the S2 File.

# 3. Results

### 3.1 Identification of plant stress response gene homologs

To identify plant defence gene homologs in silver birch and hybrid aspen, the BLAST hits from *Arabidopsis thaliana* query protein sequence were used in an additional reciprocal BLAST against all NCBI plant taxa. Candidate tree sequences with 99% coverage to any plant protein with a matching gene annotation were shortlisted by the employed BLAST algorithms. To confirm the homology, a phylogenetic analysis using all candidate homologues for a gene of interest (GOI) from hybrid aspen and silver birch were compared alongside two well annotated herbaceous plant (*Arabidopsis thaliana* and *Medicago truncatula)* homologues as well as liverwort *Merchantia polymorpha* homologue as an outgroup (Fig 1A). For each GOI, hybrid aspen and silver birch sequences with the highest coverage and identity to *A. thaliana* and *M. truncatula* and different from *M. polymorpha* were selected for primer design (S1 Table). Primer pairs displaying a single-peak melting

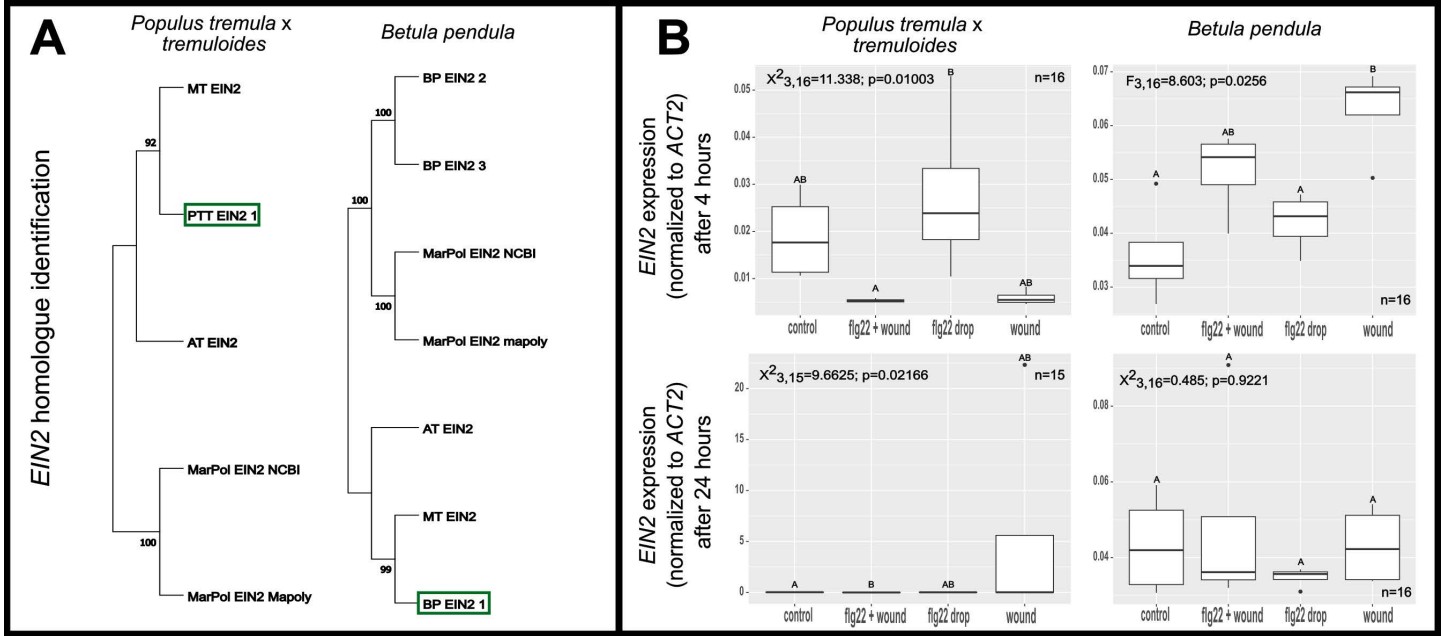

**Fig 1. Example identification of *EIN2 (ETHYLENE INSENSITIVE 2)* homologue in silver birch and hybrid aspen for downstream characterisation of systemic intra-plant biotic stress responses. (A)** Phylogenetic analysis of *EIN2* homologues from *Populus tremula* x *tremuloides (PTT)* (left) and *Betula pendula* (BP) (right). Green rectangles indicate the sequences selected for primer design. The trees are based on Maximum likelihood (bootstrap) and include the homologous sequences from *A. thaliana (*AT*)*, *M. truncatula* (MT) and *M. polymorpha* (MarPol). Phylogenetic trees for all other genes are displayed in S1 Fig (BP) and S2 Fig (PTT). **(B)** Relative expression of *EIN2* four and 24h post stimulation with *flg22*, wounding or their combination. Ct values were calculated as the average from 3 technical replicate measurements for each sample. Each boxplot represents the average of 4 biological replicates. The total number of replicates for statistical analysis across treatments is indicated in each panel. Gene expression was normalized to ACT2 as reference (ΔCt). Three test genes and ACT2 reference were measured together in the same qPCR run for 4 independent biological replicates of at least 2 treatments. The values for all gene ΔCt are available in S2 Table. Letters above boxplots indicate statistically significant differences between treatments, determined with two-tailed ANOVA or Kruskal-Wallis tests, with subsequent post-hoc tests.

curve, as well as yielding the desired amplicon (S3A and S3B Fig) were further tested for their PCR efficiency (E%) which typically ranged between 90–110% (S1 Table). *ACT2* homologues in silver birch and hybrid aspen were selected as a reference genes as they typically displayed earlier amplification (Ct = 18) compared to *TUB5*, *SAND* or *GAPDH2* (S3C Fig) as well as did not display any significant differences in the amplification cycle across water, *flg22,* wounding and wounding+*flg22* treatments in silver birch at 4h (F = 1,245; p = 0,337) and 24h (F = 0,64912; p = 0,6101) or hybrid aspen at 4h (F = 1,247; p = 0,336) and 24h (F = 1,9681; p = 0,2345). To confirm whether normalized expression results obtained with a single (*ACT2*) reference gene would differ from using multiple references, we performed a control check with random test genes and treatments using one additional reference gene candidate for each tree species (S5 Fig). Since similar results were obtained with either a single or two reference gene normalization, the downstream analysis for all samples was performed with a single *ACT2* reference at 4h and 24h after stimulation of a local leaf. Time-point and species-specific expression trends were observed such as induction of *EIN2* in distal leaves upon local wounding in silver birch or suppression in hybrid aspen.

### 3.2 Characterisation of stress responsive gene expression in systemic leaves

The tested wounding and *flg22* stimuli and their combination demonstrated significantly different effects (relative to the control) on gene expression patterns in silver birch (p < 0.01), as well as hybrid aspen (p = 0.03), indicating clear inducible

systemic responses in the distal leaves (S3 Table). The gene expression patterns showed dependency on time after stimulation of the local leaf, highlighting species specific early and late gene response to biotic stress (Fig 2). Birch showed significant early induction of *LOX2* (log$_2$ fold change = 2.98; p = 0.03), *MPK3* (log$_2$ fold change = 1.73; p < 0.01) and *EIN2* (log$_2$ fold change = 0.81; p = 0.03) 4 h after wounding. The effect, however, was clearly transitory as it disappeared after 24 h. In contrast, *flg22* treatment did not display any significant gene induction in systemic tissues neither 4 h nor 24 h after stimulation of the local leaves. Four hours after the application of the combined wounding and *flg22* treatment only *MPK3* expression was significantly elevated (log$_2$ fold change = 1.34; p = 0.03), indicating potential antagonistic responses to wounding and *flg22* treatment in silver birch.

In contrast to silver birch, hybrid aspen showed significant interaction between treatment type, identity of the activated genes and timepoints after stimulation (p < 0.001) (S3 Table), indicating complex control. Only the combined wounding and *flg22* treatment induced significant aspen gene activation at 4 h and 24 h (Fig 2). *JAZ10* was significantly (log$_2$ fold change = 5.18; p = 0.01) induced 4 h after the combined wounding and *flg22* treatment and displayed increased expression relative to the water control in the wounded plants as well, albeit not significantly. Surprisingly, the known wound-responsive JA pathway marker gene *LOX2* showed no significant induction in hybrid aspen at neither 4 h nor 24 h after the stimulation. The ethylene signalling gene *EIN2* displayed significant suppression (log$_2$ fold change = −2.03; p = 0.01) 24 h after the combined wounding and *flg22* treatment. Still, it demonstrated a tendency to be induced after wounding, suggesting potential opposing effects of *flg22* and wounding in the systemic responses of hybrid aspen leaves (Fig 2). However, without a wounding plus water control, the combined treatment does not allow clear separation of synergistic or additive interactions from potential effects of enhanced entry at the wound.

### 3.3 Promoter divergence in stress responsive genes of birch and aspen

Given the species-specific responses of the measured genes to wounding and *flg22* treatment, we hypothesised that promoter divergence may in part explain the different regulation of genes such as *LOX2* and *EIN2* in response to the same wounding or *flg22* stimulus. Alternatively, we also considered the possibility that droplet treatments containing *flg22* may have differentially penetrated the leaf surface of silver birch and hybrid aspen and perhaps contributed to the different responses to *flg22*. To explore these possibilities, we first performed additional experiments to determine droplet contact angles [45] with the leaf surface as well as measured leaf wetting by submergence in water [40] as well as rhodamine B dye to enable contrast microscopy. Interestingly, we found that leaves of both species demonstrate droplet contact angles indicative of relatively good to high surface wetting properties in birch and aspen, respectively, according to Aryal & Neuner [45] (S8 Fig). Next, we submerged the leaves for 3 h to determine water uptake from leaf surface into the apoplast and concluded that both species display comparable wetting characteristics (S8 Fig). Furthermore, the application of 5 µL rhodamine B droplet confirmed penetration of colouring into mesophyll in both aspen and birch (S8 Fig). While the saturation was relatively higher and more uniform across aspen leaves compared to patchy dye penetration across birch leaves, the stomata openings appear to serve as passage for rhodamine B penetration in birch leaves (S8 Fig), providing rhodamine entry into mesophyll. Given that *flg22* can show activity in nM concentration range [34] and we applied *flg22* at 1 µM concentration, it may be unlikely that the observed differences in gene expression between species could be attributed to differential dosage of *flg22* in the apoplast or lack of droplet uptake by the leaf surface.

To characterise the potential evolutionary divergence of tree promoter sequences that may contribute to differential binding of transcription factors to the measured gene regulatory sequences, we performed additional *in silico* analysis of silver birch and hybrid aspen *LOX2* putative promoter sequence and TCP transcription factor (TF) binding sites within 5k region upstream UTR and TSS (Fig 3A). TCP TFs are known regulators of JA synthesis in plant immunity and development [46]. *Populus tremula x tremuloides* and *Populus trichocarpa* demonstrated good synteny of conserved regions throughout the *LOX2* promoter (Fig 3B). In contrast, hybrid aspen and silver birch showed comparatively short conserved fragments (12–20nt) rearranged throughout the 5k promoter region (Fig 3C). Moreover, the conserved promoter regions

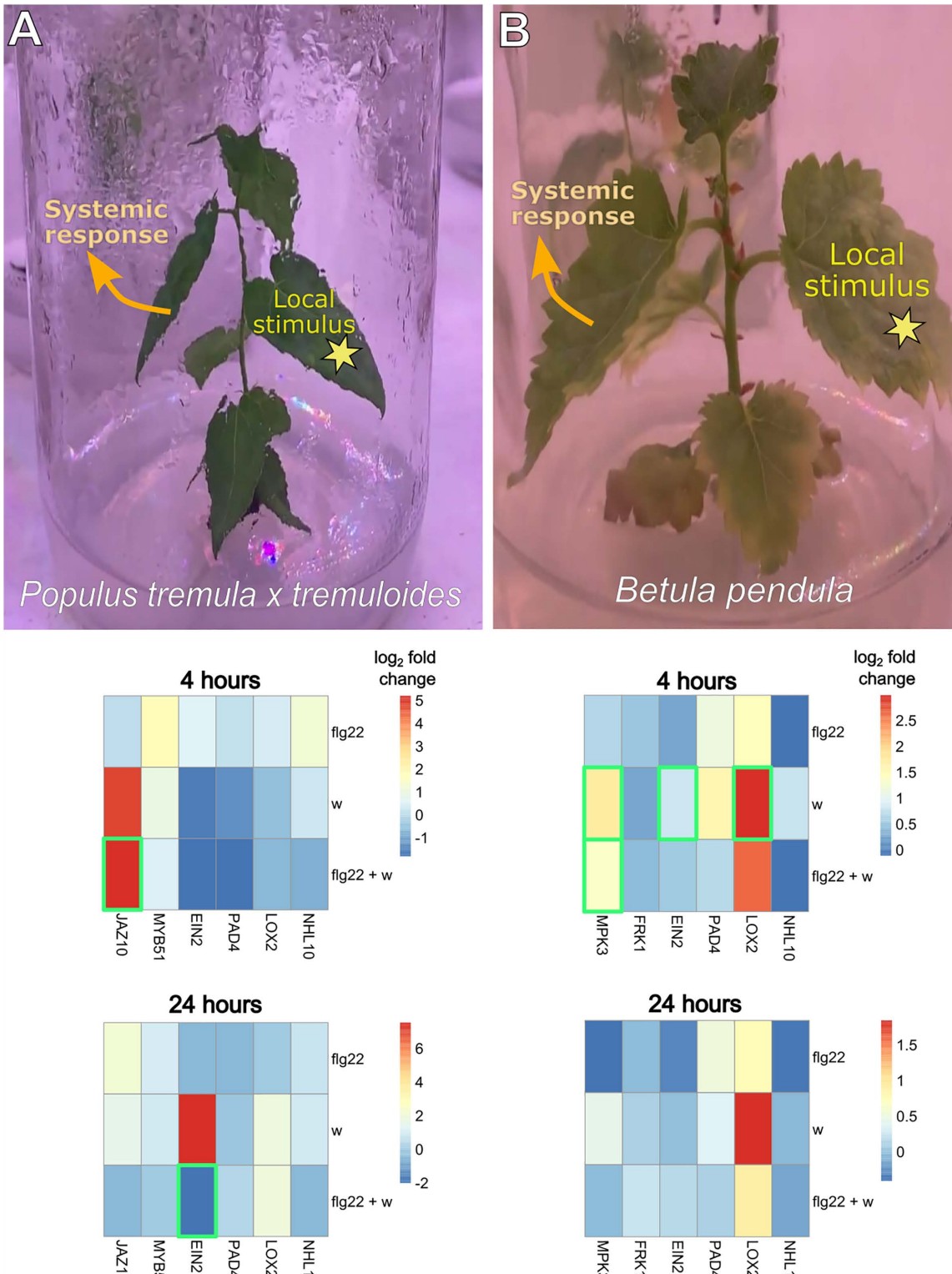

**Fig 2. Experimental setup and relative gene expression in systemic leaves of hybrid aspen *Populus tremula* x *tremuloides* (column A) and silver birch *Betula pendula* (column B) clones in response to wounding and *flg22* treatment.** The stress stimuli were applied to a local leaf and systemic responses were measured in a distal leaf of *in vitro* grown tree clones. The heatmaps demonstrate log$_2$ fold-changes in gene expression

(x-axis) for each treatment group (y-axis) relative to the water control at 4 hour and 24 hours post stimulation of a distal leaf. Each rectangle represents the average of 4 biological replicates (3 in the cases of silver birch 4h *flg22 LOX2, MPK3, NHL10*; hybrid aspen 24h *flg22 EIN2, LOX2, PAD4* and 24h *flg22+w MYB51*). *JAZ10* and *MYB51* were analysed in *P. tremula x tremuloides* but *MPK3* and *FRK1* – in *B. pendula* because the desired amplicon length and specificity was reached only in one of the species. *EIN2, PAD4, LOX2, NHL10* were analysed in both species. The green rectangles indicate statistically significant (p<0.05) differences in gene expression in systemic leaves based on ANOVA post-hoc tests. Local leaf stimuli are *flg22* − 5 µL of 1µM flagellin solution; *w* – wounding with forceps; *flg22+w* – combination of the *flg22* with wounding. The fold change is calculated using the formula $\Delta\Delta Ct=\log2(x/y)$, where x – the average normalized expression ($\Delta Ct$) of a select treatment and y – the average normalized expression ($\Delta Ct$) of the control group. $\Delta Ct$ is calculated as gene expression normalized to *ACT2* as a reference. Full normalized expression data for each gene along with baseline expression in water control treatments are displayed in S7 Fig. Source data for calculating the fold-change values is available in S2 Table.

(e.g., region 9) displayed differences in predicted TF binding sites (Fig 3C). To further verify this, we selected binding motifs of known *LOX2* transcriptional regulators – positive regulator *TCP4*, negative regulator *TCP20,* and their interactor TCX8 (Fig 3D). Since the TF binding matrices of TCP and TCX factors are only available for *Arabidopsis thaliana* homologs, we used this only as indicative proxy analysis for their putative correspondence to *LOX2* and *EIN2* from hybrid aspen and silver birch. The two genes were selected because they originally showed tree-specific differences in expression upon wounding or WF stimuli (Fig 2). As control, we also included *AtLOX2* and *AtEIN2* promoter sequences. Interestingly, TCP4 binding sites were found in both *BpLOX2* and *BpEIN2* but not in the respective hybrid aspen homologs (Fig 3E). TCP20 binding sites were found only in *PttLOX2* but not *BpLOX2* promoter (Fig 3F). Furthermore, the TCP4 and TCP20 binding motifs were found in EIN2 promoter regions in either of the two tree species but not *A. thaliana*, potentially suggesting that downstream gene responses to TCP activating signals (such as wounding) may be different in *Arabidopsis* and hybrid aspen or silver birch. TCX8 – a known interactor of class I and II TCPs and DREAM complex [47] – displayed numerous but distinct binding sites along the promoter regions of *LOX2* and *EIN2* from all three species (Fig 3G), further highlighting putative complex regulatory circuits that could underlie the species-specific responses to wounding and *flg22* based on the observed promoter divergence (Fig 3C).

## 4. Discussion

### 4.1 In vitro tree clones as models for plant systemic signalling research

The present study is the first to our knowledge to employ *in vitro* propagated tree clones as a model for studying intra-tree responses to stress signals and offers a scalable screening platform for design and discovery of tree stress response genes to diverse biotic stressors and their combinations. The applied methods provide certain potential advantages, including rapid propagation time, genetic uniformity due to clonal propagation, controlled growth and media conditions, sterile environment absent from uncontrolled pest or pathogen infections and the ability to precisely subject trees to selected defence elicitors or individual stress elements, potentially overcoming many challenges in future elucidation of molecular mechanisms for tree biotic and abiotic stress tolerance associated with heterogeneity and interaction of multiple environmental and stress factors in greenhouse and field studies [21,24]. Early identification of stress-responsive genes using *in vitro* systems would accelerate follow-up functional studies in soil-grown saplings under growth chamber, greenhouse, or field conditions, enabling a wide range of biological investigations.

The experimental system herein allowed for optimal selection of tree stress inducing treatments and their response time estimation. For example, we discovered that only the combined treatment of wounding and *flg22* application to local leaves triggered significant systemic activation of JA-response gene *JAZ10* after 4h in hybrid aspen (Fig 2), while the individual wounding and *flg22* treatments failed to induce consistent early responses. In contrast, leaf wounding displayed more early responding genes compared to the other treatments in silver birch. Selection of reliable and reproducible stress inducing treatments and their marker genes could be a key primer for several applications such as mechanistic studies on inter-tree signals and responses [Orlovskis *et al*., 2024] in the future.

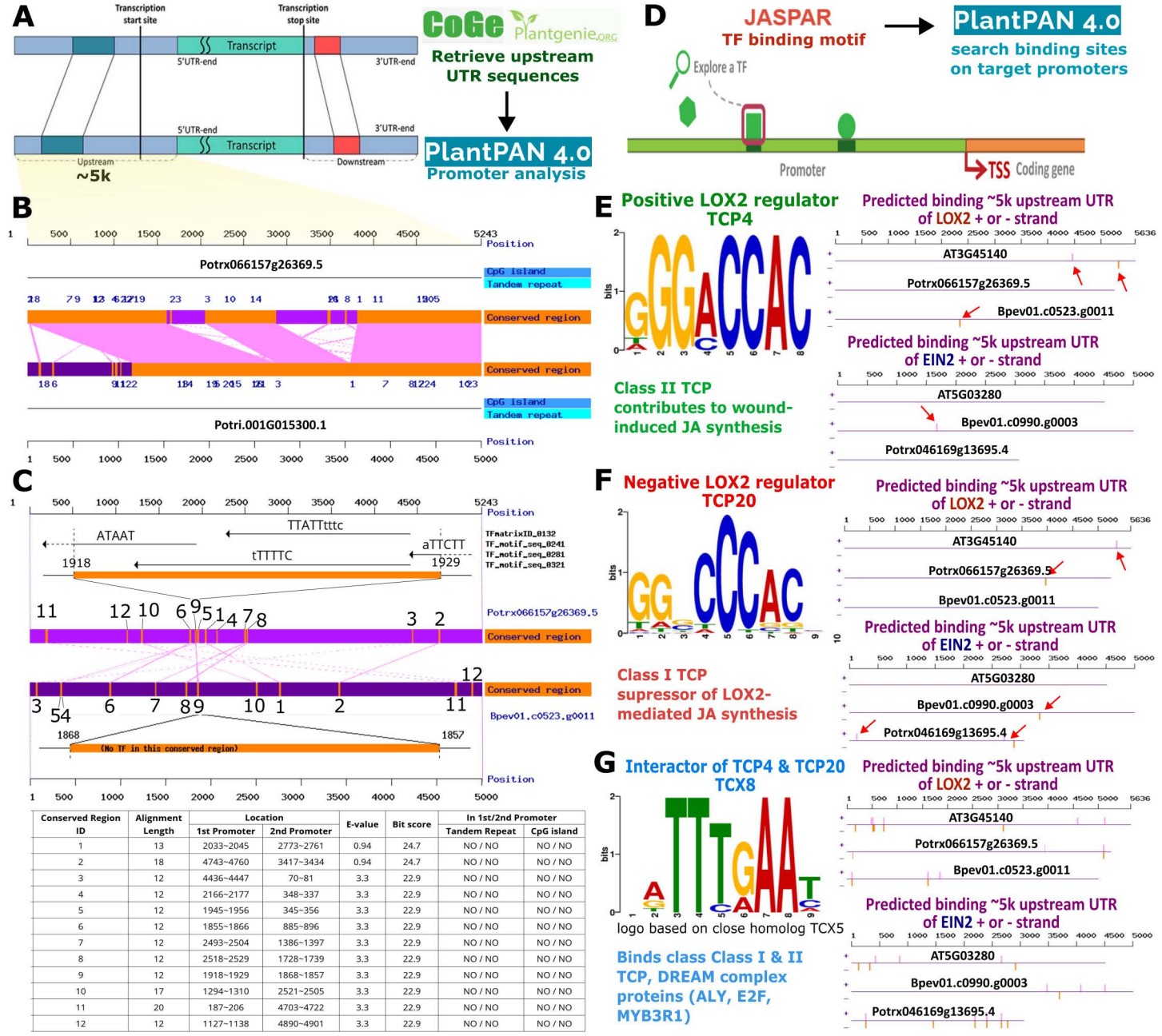

**Fig 3. Silver birch and hybrid aspen display sequence divergence in *LOX2* promoter region and putative transcription factor binding sites.** 5k upstream UTR sequences of the *LOX2* gene were retrieved from the CoGe and PlantGenIE databases and aligned with PlantPan v4.0 to identify conserved regions and transcription factor (TF) binding motifs in the putative promoter region of the *LOX2* gene **(A)**. Promoter regions of the *LOX2* homologue from *Populus tremula x tremuloides* (Ptt, Potrx066157g26369.5) displays higher sequence conservation and synteny with *Populus trichocarpa* (Pt, Potri.001G015300.1) **(B)** compared to more divergent promoter and putative TF binding sites in *Betula pendula* (Bp, Bpev01.c0523.g0011) **(C)**. Only 12 conserved regions of 12-20 nt in length were identified in the 5k upstream UTR region of *Bp* and *Ptt* corresponding to different locations within the putative promoter region. As an example, region 9 from *Ptt* and *Bp* displays different TF binding sites from the available *PlantPan* annotations **(C)**. Binding matrices of known *LOX2* regulators – TCP and TCX transcription factors – were acquired from the JASPR database for their specific correspondence to *LOX2* and *EIN2* promoter regions from *Arabidopsis thaliana* (At), *Ptt* and *Bp* using PlantPan v4.0 promoter analysis tool **(D)**. TCP4, TCP20 and TCX8 binding sites are displayed on the + and − strand of *LOX2* and *EIN2* promoter regions (red arrows) and compared across the three plant species **(E-G)**. Only the 3k upstream UTR region of *EIN2* homologue from *Ptt* (*Potrx046169g13695.4*) was retrieved due to close proximity of the transcription stop site of the upstream gene.

## 4.2 Species specificity of SA or JA responses upon wounding or *flg22* application

While certain marker genes of early responses in pattern triggered immunity (PTI) have been well described in *Arabidopsis* such as early response genes *NHL10 (NDR1/HIN1-LIKE 10)* or MAPK-specific target gene *FRK1 (FLG22-INDUCED RECEPTOR KINASE1)* [48], no significant gene expression differences compared to the water control treatment were observed for these genes in *flg22* treatment in silver birch or hybrid aspen, potentially indicating different timing or species specific differences in the mechanisms of early FLS2-mediated PTI signalling and their responses. These can be attributed to species-specific differences in the recognition of shortened *flg22* ligands by FLS2 receptor- co-receptor complexes [49] or, alternatively, heterogeneity of flagellin fragments [50] produced by leaf endophyte communities that may interfere with responses to exogenously applied pathogenic flagellin.

Alternative explanation for species-specific effects would be differential penetration of *flg22* through the cuticle and leaf surface of hybrid aspen and silver birch. While hybrid aspen leaves displayed greater wettability than silver birch, the water and dye solution uptake by leaves during the 3h period was comparable in both species and not enhanced by the addition of a surfactant (S8 Fig), corroborating the capability of foliar solute uptake in trees [51]. Furthermore, application of 5 µL dye droplet comparable to the *flg22* treatment suggests more uniform uptake across the leaf surface of aspen but more patchy entry into the apoplast via open stomata in silver birch. While we did not determine the penetration of water or *flg22* via the wound sites in comparison to passive diffusion across leaf surface, inclusion of water + wounding control could also test for the possibility of dilution of DAMPs (e.g., cAMP, cellobiose) by the added 5 µL solution. Thus, future experiments may better distinguish whether interplay between wound and *flg22*-induced signals may be associated with downstream hormonal crosstalk or differences in elicitor concentration and entry into the apoplast.

*PAD4 (PHYTOALEXIN DEFICIENT4)* and *EDS1 (ENHANCED DISEASE SUSCEPTIBILITY1)* constitute a well-established module in PTI [52], as well as abiotic stress [53] signalling and a positive regulator of salicylic acid (SA) pathway in *Arabidopsis* [54]. However, *PAD4* did not display significant response to *flg22* or combined wounding and *flg22* treatments of the tree species, suggesting potentially different *flg22* perception and signalling mechanisms than *Arabidopsis*. Both *PAD4* and plant transcription factor *MYB51* are also involved in glucosinolate and callose metabolism in innate immune responses to *flg22* [55,56] and mediate plant responses to phloem feeding insects [57,58]. When tested in hybrid aspen, *MYB51* did not show any significant responses to wounding or *flg22* treatments.

*LOX2 (LIPOXIGENASE2)* encodes a key enzyme in jasmonic acid (JA) biosynthesis [59] while *JAZ10 (JASMONATE ZIM-Domain10)* is an important component in downstream JA signalling [60] in wound responses and control of glucosinolate production in *Arabidopsis* [28]. Interestingly, *LOX2* displayed induction in silver birch but not in hybrid aspen (Fig 1), indicating species-specific responses or response timing to wounding. However, *JAZ10* displayed upregulation upon wounding as well as the combined wounding and *flg22* treatment in aspen. Since the JA and SA pathways are considered antagonistic [61], this may potentially illustrate a more complex interaction between SA and JA pathways in hybrid aspen compared to *Arabidopsis*.

*MAPK (MITOGEN ACTIVATED PROTEIN KINASES)* are important regulators of herbivory associated local wounding as well as plant systemic responses [62]. MPK3 was induced in systemic tissues upon distal wounding and *flg22* treatment in silver birch (Fig 2). *MPK3* and *MPK6* are known activators of ethylene (ET) biosynthesis in *Arabidopsis* [63,64]. As reviewed in Broekgaarden et al. [64], ET signalling also regulates *flg22*–triggered PTI and hormonal crosstalk between SA-mediated responses to biotrophic pathogens and JA-mediated responses to wounding and chewing herbivores. *ETHYLENE INSENSITIVE2 (EIN2)* is an important ET signalling component [65] and was induced in silver birch upon wounding, consistent with the observed *LOX2* and *MPK3* responses and involvement in wound signalling [64].

Interestingly, the opposite regulation of *EIN2* during wounding alone versus combined wounding and *flg22* treatment in hybrid aspen may potentially indicate crosstalk between the *flg22*-dependent SA pathway and the wound-inducible JA pathway. Moreover, the culture medium used in our experiments contained 0.5 µM IAA (auxin), which is required for rooting and propagation of clonal cuttings but is also known to interact with other phytohormone pathways, including

suppression of SA-mediated host defences and promotion of bacterial virulence [26,66,67]. Consequently, systemic stress responses observed under *in vitro* conditions may differ in soil-grown plants and warrant further investigation under more natural growth substrates.

Finally, as this study focused on a single commercially grown, high-yielding clone per species, future work should assess stress-responsive gene expression across a broader range of genotypes. Early identification of such markers using *in vitro* systems would facilitate targeted functional validation in soil-grown saplings under growth chamber, greenhouse, and field conditions, thereby supporting diverse downstream biological investigations.

## 5. Conclusion

We have identified *BpLOX2, BpMPK3*, and *BpEIN2* as suitable early response marker genes for wounding in silver birch, while *PttJAZ10* and *PttEIN2* responded to a combination of wounding and flagellin treatment in hybrid aspen. The selection of biotic stress treatments and the identification of response marker genes will contribute to future research on systemic stress responses both within and between trees, using *in vitro* and soil-propagated clones. This system offers a valuable platform for exploring the role of different microbial inoculants and the cross-communication between tree species or intra-specific genotypes. Specifically, it will facilitate investigations into the impact of pathogen attacks on trees interconnected by common mycorrhizal networks (CMN), helping to uncover the mechanisms behind CMN-mediated defence responses in forest ecosystems.

## Supporting information

**S1 Table. List of silver birch and hybrid aspen stress response genes used in the study.** Primer sequences are provided alongside the corresponding gene ID from TAIR, CoGe, PlantGenIE databases. Amplification factor was calculated based on 10-fold template DNA dilution series ranging from 100ag to 1ng and displayed for all genes used in Fig 1. Genes that did not display a specific single band in the electrophoresis and did not produce consistent melting curves (S3 Fig), the amplification factor was not calculated.
(XLSX)

**S2 Table. Source data table (.xls) with ΔCt and ΔΔCt values used to generate expression graphs in Fig 1B, heatmaps in Fig 2 and expression boxplots in S5 Fig.** The expression of each test gene was normalized by the expression of reference gene (ACT) using the formula $\Delta Ct = Eref^{Ctref}/ Etest^{Cttest}$, where Eref – primer efficiency of reference gene, Etest – primer efficiency of test gene, taken to the power (^) of Ctref – Ct value of reference gene, Cttest – Ct value of the test gene, respectively. log2 fold change was calculated using the formula $\Delta\Delta Ct = log2(x/y)$, where x – the average normalized expression (ΔCt) of a select treatment and y – the average normalized expression (ΔCt) of the control group.
(XLSX)

**S3 Table. Generalized linear model results for testing the differences in gene expression between treatments across the panel of genes used in Fig 1.** The model tests for the interaction of wound or *flg22* treatments with 4h and 24h timepoints and individual expression patterns of each gene (Fold Change ~ Treatment * GeneID * Time-point).
(XLSX)

**S1 Fig. Cladograms of putative *Populus tremula x tremuloides* marker gene homologs.** Phylogeny was based on translated protein sequences of *JAZ10, MYB51, EIN2, PAD4, LOX2, NHL10, FRK1, MPK3*, and reference gene *ACT2* from *Populus tremula x tremuloides* (PTT), *Arabidopsis thaliana* (AT), *Medicago truncatula* (MT), *Marchantia polymorpha* (MarPol). In case of multiple spliced variants, all sequences were included for tree construction using the Maximum likelihood method. Green box denotes the homolog used for primer design in the study.
(TIFF)

**S2 Fig. Cladograms of putative *Betula pendula* marker gene homologs.** Phylogeny was based on translated protein sequences of *JAZ10, MYB51, EIN2, PAD4, LOX2, NHL10, FRK1, MPK3*, and reference gene *ACT2* from *Betula pendula* (BP), *Arabidopsis thaliana* (AT), *Medicago truncatula* (MT), *Marchantia polymorpha* (MarPol). In case of multiple spliced variants, all sequences were included for tree construction using the Maximum likelihood method. Green box denotes the homolog used for primer design in the study. In cases where multiple potential homologues clustered near MT and/or AT sequences, the candidate with the highest homology scores (provided by the BLAST tools used in each case) as well as with similar results in the reverse BLAST procedure were chosen.
(TIFF)

**S3 Fig. Melting curves for all tested gene amplicons in rt-qPCR data from representative samples of *Betula pendula* and *Populus tremula x tremuloides*.**
(TIFF)

**S4 Fig. Electrophoresis gels of stress response gene amplicons in *Betula pendula* (A) and *Populus tremula x tremuloides* (B) as well as gene amplification curves (C) and variance across treatments (D) for tree reference gene selection.** 35-cycle qPCR amplicons were used to check for primer specificity, target size and absence of double bands **(A-B)**. deltaRn indicates the intensity of SYBR signal corresponding to target amplification during 35 qPCR cycles. Reference gene *ACT2* was selected based on earlier amplification and uniform expression across the different stress treatments in the qPCR amplification curve **(C)**.
(TIFF)

**S5 Fig. Calculation of primer efficiency for genes used in this study. (A)** All stress marker genes used in this study are listed with their amplification efficiency (E%) and amplification factor (E) used for calculating ΔCt values. Calculation of E% is based on formula $E(\%) = \left( \frac{-1}{10^{\text{Slope}}} - 1 \right) \times 100$. **(B)** Slope values are based on the regression lines for the Ct values for amplification of 10-fold dilution series of the template DNA, starting from 100 pg.
(PNG)

**S6 Fig. Reference gene testing for *B. pendula* (A) and *P. tremula x tremuloides* (B).** To test whether *ACT2* is the optimal reference gene for defence gene expression testing, qPCR was carried out with an additional potential reference gene - *GAPDH2* and *TUB5* for *B. pendula* and *P. tremula* x *tremuloides* respectively and the relative expression of select genes and treatments was calculated relative to the geometric mean of both potential reference genes and compared to original expression data (only *ACT2* as a reference). Geometric mean of two selected reference genes is calculated with the formula $\frac{E_{ref1}^{Ct(ref1)} \times E_{ref2}^{Ct(ref2)}}{E_{test}^{Ct(test)}}$, where $E_{ref}$ – primer efficiency of reference genes 1 and 2, $E_{test}$ – primer efficiency of test gene, *Ct(ref)* – Ct value of reference genes 1 and 2, *Ct(test)* – Ct value of the test gene. While both tested hybrid aspen genes showed differences in statistical significance, the expression tendencies remained similar in both cases. Difference in result significance could potentially be explained by the fact that the original models, which included all treatments, used ANOVA and/or Kruskal-Wallis tests, while the present experiment used t and Wilcoxon signed ranked tests, since only two groups were compared.
(TIFF)

**S7 Fig. Marker gene expression relative to reference gene in *Betula pendula* (A) and *Populus tremula x tremuloides* (B).** Boxplots represent the interquartile range (IQR), the line indicates the median expression, the whiskers – variance within 1.5x IQR. Summary results of ANOVA or Kruskal-Wallis and corresponding post-hoc tests are provided for each marker gene. Gene expression was normalized to ACT2 as reference. Ct were calculated as average from 3 technical replicate measurements for each sample. Test and reference gene was analyzed on the same qPCR plate. Each plate contained 4 independent biological replicates of at least 2 treatments. Sample size (n) for ANOVA or Kruskal-Wallis is shown at the bottom-left corner of each graph. Cases where sample size was 15 instead of 16, were related to insufficient cDNA amount.
(PNG)

**S8 Fig. Leaf Summary of wetting experiments.** (**A**) Microscopy of *B. pendula* and *P. tremula x tremuloides* leaves treated with rhodamine B. Three different treatments were used – submergence in rhodamine B solution (Rb), submergence in rhodamine B solution with added Tween 80 (Rb + Tween 80), a single drop (5 µL) of rhodamine B solution on the abaxial leaf surface (Rb drop) and a control group of a leaf submerged in water (H2O). Submergence photos were taken with 10x magnification; Rb drop photos with 40x. All photos were taken 3 hours post treatment. White triangle on the *P. tremula x tremuloides* Rb drop photo indicates a stoma – a potential entry point for Rb solution (and water) infiltration. B – Difference in water uptake 3 hours post treatment between different treatments and both tree species. No significant difference was observed between different treatments, while *B. pendula* water uptake was significantly higher than *P. tremula x tremuloides*. Regardless, all experimental groups demonstrated water uptake through the leaf surface. C – results of the leaf surface wettability experiment. The wettability of the leaf surface for both tree species was carried out, by determining the surface contact angle of a water drop on the surface of a leaf (shown in the photographs). The median value of the surface contact angle of the water droplet was lower than 90 degrees, indicating the wettability of the leaf surface for both species.
(TIFF)

**S1 File. Detailed description of the modified CTAB extraction protocol used for total RNA extraction.**
(DOCX)

**S2 File. A representative selection of R studio code lines used in data analysis.**
(DOCX)

## Author contributions

**Conceptualization:** Zigmunds Orlovskis.

**Data curation:** Kārlis Blūms, Roberts Matisons, Zigmunds Orlovskis.

**Formal analysis:** Kārlis Blūms, Roberts Matisons, Zigmunds Orlovskis.

**Funding acquisition:** Zigmunds Orlovskis.

**Investigation:** Kārlis Blūms, Baiba Krivmane, Maryna Ramanenka, Zigmunds Orlovskis.

**Methodology:** Kārlis Blūms, Baiba Krivmane, Maryna Ramanenka, Dainis Ruņģis, Zigmunds Orlovskis.

**Project administration:** Mārtiņš Zeps, Zigmunds Orlovskis.

**Resources:** Roberts Matisons, Dainis Ruņģis, Mārtiņš Zeps, Zigmunds Orlovskis.

**Supervision:** Dainis Ruņģis, Mārtiņš Zeps, Zigmunds Orlovskis.

**Validation:** Kārlis Blūms, Roberts Matisons, Zigmunds Orlovskis.

**Visualization:** Kārlis Blūms, Zigmunds Orlovskis.

**Writing – original draft:** Kārlis Blūms, Zigmunds Orlovskis.

**Writing – review & editing:** Kārlis Blūms, Baiba Krivmane, Roberts Matisons, Dainis Ruņģis, Zigmunds Orlovskis.

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
