## [Decision Letter · Decision Letter 0]

20 Oct 2025

Dear Dr. Orlovskis,

We look forward to receiving your revised manuscript.

Kind regards,

Benedicte Riber Albrectsen

Academic Editor

PLOS ONE

“This work was supported by Latvian Council of Science Fundamental and Applied Research grant lzp-2022/1-0283.”

“This work was supported by Latvian Council of Science Fundamental and Applied Research grant lzp-2022/1-0283.”

“This work was supported by Latvian Council of Science Fundamental and Applied Research grant lzp-2022/1-0283.”

4. We notice that your supplementary tables are included in the manuscript file. Please remove them and upload them with the file type 'Supporting Information'. Please ensure that each Supporting Information file has a legend listed in the manuscript after the references list.

Reviewers' comments:

Reviewer's Responses to Questions

**Comments to the Author**

1. Is the manuscript technically sound, and do the data support the conclusions?

Reviewer #1: Yes

Reviewer #2: Partly

2. Has the statistical analysis been performed appropriately and rigorously?

Reviewer #1: Yes

Reviewer #2: I Don't Know

3. Have the authors made all data underlying the findings in their manuscript fully available?

Reviewer #1: Yes

Reviewer #2: No

4. Is the manuscript presented in an intelligible fashion and written in standard English?

Reviewer #1: Yes

Reviewer #2: Yes

Reviewer #1: General comments:

This study presents an in vitro methodological approach to study systemic signalling and stress responses in hybrid This manuscript provides a novel and well-described in vitro approach to investigate systemic signaling in deciduous trees, identifying species-specific marker genes for wounding and flagellin-induced responses. The work is methodologically valuable and offers a foundation for future research in tree stress biology. However, it would benefit from greater integration of physiological validation, effect size reporting, and concise presentation to enhance its impact.

The in vitro clonal propagation and stimulation system for woody species is a valuable and scalable model for controlled stress signaling experiments rarely demonstrated in trees. Comparing Betula pendula and Populus tremula × tremuloides adds ecological and evolutionary relevance, showing species-specific systemic signalling patterns.

Measuring responses in distal leaves (not locally treated ones) strengthens the work’s link to systemic signaling, rather than local wound responses. Combining in silico gene homologue discovery, phylogenetic confirmation, and experimental validation adds credibility to the choice of marker genes. Detailed description of plant material, treatments, RNA extraction, and qPCR methodology allows reproducibility for methodological transparency. Its very nice work. All the very best to authors.

Accepted with minor revision.

Specific comments:

I have only few comments-

1. Abstract nicely written, but could include one key result (e.g., “LOX2, MPK3, and EIN2 were early wounding markers in birch; JAZ10 and EIN2 responded in aspen to combined treatments”) for a stronger takeaway.

2. Consider separating ecological background from molecular aims for clarity.

3. Methodology is very detailed and good for reproducibility. Could move the long RNA extraction protocol to Supplementary Methods and summarize the key steps in the main text. While the methodological development is valuable, the study mainly reports expression differences without deeper mechanistic exploration (e.g., hormone quantification, protein activity, imaging of ROS).

4. Include one representative tree figure in the main text to visually confirm gene homology.

5. Figure 1: It’s unclear how fold changes were normalized (ΔΔCt formula not shown in text). Add a legend clarifying “relative to ACT2 and to control.”

6. Expand discussion on why LOX2 was inactive in aspen but active in birch, possibly due to promoter divergence or JA pathway suppression.

7. Conclusion is well stated. You could strengthen it with one or two sentences about future applications, e.g., “The system can now be used to test microbial inoculants or cross-communication between tree genotypes.”

Reviewer #2: The authors use in vitro propagated clones of hybrid aspen and silver birch to profile systemic transcriptional responses at 4 h and 24 h after local wounding, flg22, or the combination. They identify homologs of eight Arabidopsis defense related genes, design primers, and run RT-qPCR with ACT2 as the reference. In birch, wounding induces LOX2, MPK3, and EIN2 at 4 h. In aspen, the combined treatment induces JAZ10 at 4 h and suppresses EIN2 at 24 h. The study proposes an in vitro screening workflow for marker nomination in forestry species.

General assessment

The in vitro clonal approach for systemic signaling is practical and could be useful for early marker screening before greenhouse or field work. Several parts need strengthening for clarity and rigor, including alignment of aims and conclusions, control design, stimulus delivery and timing, reference gene normalization, figure set, and data transparency.

Major comments

1. Abstract states an aim to identify and validate known plant defense markers in aspen and birch, while Conclusion presents a methodological pipeline for homologue discovery. Please harmonize Abstract, Introduction, and Conclusion so the study goal and take-home message are consistent across sections.

2. The combined treatment is wound plus 5 uL of 1 uM flg22 applied on the wound, while the control is water on an unwounded leaf. A wounded plus water control is needed to isolate the added contribution of flg22 on a wound. If these data are not available, state this limitation clearly and moderate any claim of additivity or synergy.

3. FLS2 perceives flg22 in the apoplast at the plasma membrane. A small surface droplet can have variable penetration through cuticle and stomata in in vitro leaves, which can explain weak flg22 alone effects, and wounding can enhance entry. Please justify the chosen delivery and either show that surface application gives reliable uptake such as a simple tracer or wetting validation and residence time, or consider a more reliable delivery such as gentle syringe infiltration in water or a validated spray with a minimal wetting agent, with matching mock controls. In all cases, include a wounded plus water control to isolate the added contribution of flg22. 1 uM is plausible yet responses are dose dependent and species dependent. If feasible, consider adding dose–response data, otherwise acknowledge this limitation. Early PTI peaks often occur within minutes up to about two hours. Please justify the focus on 4 h and 24 h and discuss likely missed early responses, and consider adding an early time point. Please provide a short rationale for using the combined stimulus that explains ecological co-occurrence and expected hormonal crosstalk, and please clarify whether the interaction was a planned hypothesis or exploratory. In the analysis, please report simple effects that distinguish combination effects from potential entry effects at the wound, for example, adding flg22 at a wound versus wounded plus water.

4. ACT2 was chosen because it was amplified earlier and looked uniform compared with TUB, SAND, and GAPDH. This alone does not establish stability. Evaluate at least two candidate reference genes with a standard stability method and either use two gene normalization or show ACT2 stability with no treatment effect on Ct in the same conditions. Reference gene stability can be demonstrated by flat ACT2 Ct across water, flg22, wound, and combination within each time point. Also, state clearly in the main text and in figure legends the replication structure.

5. The expression boxplots in Figure S4 show distributions and post hoc groups and thus improve interpretability. Promote them to a main figure next to the heatmap. Add n on each panel.

6. IBA is present in culture. Auxin can influence gene expression. Please justify its inclusion and discuss any potential confounding of defense marker dynamics.

7. Consider avoiding phrases like strongest response or good or useful or strong marker. Please state the measured change for each claim with its interval and adjusted p value. Please define a simple criterion for using the word marker and apply it consistently.

8. Only one clone per species was tested. Frame results as clone level patterns and putative markers to be validated across more genotypes and growth conditions including soil grown plants. Revise the Conclusion accordingly.

9. Add the following to Supporting Information. Include the exact amplicon length for each primer pair. Report the standard curve slope and R squared used to calculate amplification efficiency together with the dilution range already stated. Provide raw Ct or delta Ct tables for all samples. Provide basic RNA integrity information such as RIN values or gel traces. Deposit the R code used for data processing, statistics, and figure generation in a public repository with a persistent identifier.

Minor comments

- Fix typos and inconsistencies. Chinchilla spelling. LOX2 not LOC2. CoGe not GOGE. Remove the duplicate Louis and Shah 2015 reference.

- Phylogeny figures need higher legibility. Export higher dpi or vector graphics. Add a short note on how the final homolog was chosen when several paralogs cluster near Arabidopsis or Medicago.

- Regarding primer specificity, include representative melt curves for each target in Supporting Information and annotate band sizes on gels in Figure S3 or list the sizes in Table S1.

- The study focuses on eight genes including LOX2 and JAZ10 for wound and JA signaling, and MPK3 and EIN2 for cross talk. Please explain how the candidate list was narrowed to these eight and why other widely used wound markers were not included such as AOS, AOC, OPR3, VSP1 or VSP2, COI1, JAR1, MYC2, or protease inhibitor genes. If additional homologues were screened in silico, list them in Supporting Information with brief reasons for exclusion. This will help readers understand the design scope and calibrate conclusions.

**Do you want your identity to be public for this peer review?** For information about this choice, including consent withdrawal, please see our Privacy Policy

Reviewer #1: No

Reviewer #2: No

---

## [Author Response · Author response to Decision Letter 1]

19 Dec 2025

Point-by-point responses to reviewers are included in Response to reviewers file.

---

## [Decision Letter · Decision Letter 1]

22 Jan 2026

Dear Dr. Orlovskis,

We look forward to receiving your revised manuscript.

Kind regards,

Benedicte Riber Albrectsen

Academic Editor

PLOS One

Journal Requirements:

**Additional Editor Comments:**

The revised version should consider the requests raised by reviewer 2:

Please add one clear sentence in Results or Discussion that the absence of a wounded plus water control means the combined treatment cannot distinguish synergy or additivity from enhanced entry at the wound. Keep any language about interaction effects cautious.

2. Circadian timing note

Please add one sentence in Discussion to state whether time of day was controlled or recorded. If not controlled, note that diurnal variation may contribute to some differences.

3. Standard curve reporting

You report slopes for the qPCR dilution curves. Please add the corresponding R squared values for each assay in the supplement or state the range across assays. This helps readers judge amplification efficiency quality.

Minor comments

– If space allows, indicate n on main figure panels that show distributions. If not, the legend statement on replication is acceptable.

Reviewers' comments:

**Comments to the Author**

Reviewer #1: All comments have been addressed

Reviewer #2: (No Response)

2. Is the manuscript technically sound, and do the data support the conclusions?

Reviewer #1: Yes

Reviewer #2: Partly

3. Has the statistical analysis been performed appropriately and rigorously?

Reviewer #1: Yes

Reviewer #2: Yes

4. Have the authors made all data underlying the findings in their manuscript fully available?

Reviewer #1: Yes

Reviewer #2: Yes

5. Is the manuscript presented in an intelligible fashion and written in standard English?

Reviewer #1: Yes

Reviewer #2: Yes

Reviewer #1: Accepted. Congratulations to authors for their great work. Authors has addressed all comments very nicely.

Reviewer #2: The authors revised the manuscript on systemic transcriptional responses in distal leaves of hybrid aspen and silver birch at 4 h and 24 h after wounding, flg22, or the combination. They added tracer data to support droplet uptake, provided source tables and code, and presented checks for ACT2 stability. The overall clarity and transparency improved.

General assessment

The study now fits the scope of PLOS ONE provided a few remaining points are addressed. The design and methods are largely sound, limitations are more clearly discussed, and the new additions strengthen the technical foundation. I recommend minor revision focused on concise clarifications and one small reporting addition.

Major comments

1. Combined treatment limitation

Please add one clear sentence in Results or Discussion that the absence of a wounded plus water control means the combined treatment cannot distinguish synergy or additivity from enhanced entry at the wound. Keep any language about interaction effects cautious.

2. Circadian timing note

Please add one sentence in Discussion to state whether time of day was controlled or recorded. If not controlled, note that diurnal variation may contribute to some differences.

3. Standard curve reporting

You report slopes for the qPCR dilution curves. Please add the corresponding R squared values for each assay in the supplement or state the range across assays. This helps readers judge amplification efficiency quality.

Minor comments

– If space allows, indicate n on main figure panels that show distributions. If not, the legend statement on replication is acceptable.

**Do you want your identity to be public for this peer review?** For information about this choice, including consent withdrawal, please see our Privacy Policy

Reviewer #1: No

Reviewer #2: No

---

## [Author Response · Author response to Decision Letter 2]

4 Feb 2026

PONE-D-25-43089

Species specific marker genes for systemic defence and stress responses to leaf wounding and flagellin stimuli in hybrid aspen and silver birch

Dear editor,

We are grateful for careful consideration and evaluation of our revised manuscript PONE-D-25-43089 for publication in PlosOne. Following the feedback and comments from the reviewers and editor, we here resubmit our revised manuscript with the following corrections to our manuscript:

1. Combined treatment limitation

Please add one clear sentence in Results or Discussion that the absence of a wounded plus water control means the combined treatment cannot distinguish synergy or additivity from enhanced entry at the wound. Keep any language about interaction effects cautious.

In line 330-332 we added a new sentence to explicitly state ‘’However, without a wound plus water control, the combined treatment does not allow clear separation of synergistic or additive interactions from potential effects of enhanced entry at the wound.’’ This is corroborated by further statements in discussion (lines 471-477) from the first round of revisions, acknowledging the benefits from including the water + wound control in future experiments.

2. Circadian timing note

Please add one sentence in Discussion to state whether time of day was controlled or recorded. If not controlled, note that diurnal variation may contribute to some differences.

We thank reviewer for suggesting to include the missing detail. We performed leaf stimulation at the same time of the day for all treatments. Therefore, we included a new sentence in the methods section: (line 140-141) ‘’To minimize potential diurnal variation in gene expression, all stimuli were applied at noon (12:00), 4h after the onset of grow chamber lighting. ‘’

3. Standard curve reporting

You report slopes for the qPCR dilution curves. Please add the corresponding R squared values for each assay in the supplement or state the range across assays. This helps readers judge amplification efficiency quality.

We have added the R-squared values and the regression equations for each dilution in Supplemental Figure 5.

Minor comments

– If space allows, indicate n on main figure panels that show distributions. If not, the legend statement on replication is acceptable.

We have included the number of replicates in Figure 1. We added an extra sentence ‘’Each boxplot represents the average of 4 biological replicates. The total number of replicates for statistical analysis across treatments is indicated in each panel. ‘’ in the legend. We added a sentence in the legend of Figure 2: “Each rectangle represents the average of 4 biological replicates (3 in the cases of silver birch 4h flg22 LOX2, MPK3, NHL10; hybrid aspen 24h flg22 EIN2, LOX2, PAD4 and 24h flg22+w MYB51).”

For the ease of tracking changes in the manuscript, we include the file xxx_track changes.docx. The final version with all track changes accepted is also enclosed and uploaded to the online submission system.

We hope that the improvements presented herein will be sufficient to address the points raised by the reviewer and provide readers with new in-depth information for better interpreting the results presented herein. We are looking forward to your final decision.

Sincerely,

Dr. Zigmunds Orlovskis

---

## [Editor Report · Decision Letter 2]

11 Feb 2026

Dear Dr. Orlovskis,

Please make sure to cite online resources used for the manuscript. E.g. PlantGenIE (Sundell et. al. 2015. The Plant Genome Integrative Explorer Resource: PlantGenIE.org. The Plant Journal, 84(4), 617–632) and PlantPAN (Chow, C. N., Zheng, H. Q., Wu, N. Y., Chien, C. H., Huang, H. D., Lee, T. Y., & Chang, W. C. (2019). PlantPAN 3.0: a new and updated resource for reconstructing transcriptional regulatory networks from ChIP-seq experiments in plants. Nucleic Acids Research, 47(D1), D1155–D1163. https://doi.org/10.1093/nar/gky1081)

Please also make sure to align figures so that the same order of model system (Birch and hybrid-aspen) occur in the figure and in the legend - this is relevant for figures 1 and 2.

In addition, in Figure 1 for EIN2 in the hybrid 24h - the authors should carefully review p-value and post hoc letters ... one or the other imust be incorrect as there are only A, and ABs and no Bs in the boxplot.

Similarily, in Figure 2 in which the problem re-occurs with differnet order of birch and poplar mentioned in the legend and presented in the figure. In addition, the heat maps follow different rankings so neither treatment nor gene occur in the same order in the heatmap clusters. This difference in orders should be emphasised in the figure legend, or alternatively the heatmaps could be reorganised with fixed ranking of treatment and gene.

We look forward to receiving your revised manuscript.

Kind regards,

Benedicte Riber Albrectsen

Academic Editor

PLOS One

---

## [Author Response · Author response to Decision Letter 3]

16 Feb 2026

We are grateful for careful consideration and evaluation of our revised manuscript PONE-D-25-43089R2 for publication in PlosOne. Following the feedback and comments from the reviewers and editor, we here resubmit our revised manuscript with the following corrections in response to the requested changes:

1. Include references to online tools and resources.

We have included the missing references in materials & methods section and the reference list:

JASPAR

Baydar Ovek D, Rauluseviciute I, Aronsen DR, Blanc-Mathieu R, Bonthuis I, de Beukelaer H, Ferenc K, Jegou A, Kumar V, Lemma RB, Lucas J, Pochon M, Yun CM, Ramalingam V, Deshpande SS, Patel A, Marinov GK, Wang AT, Aguirre A, Castro-Mondragon JA, Baranasic D, Chèneby J, Gundersen S, Johansen M, Khan A, Kuijjer ML, Hovig E, Lenhard B, Sandelin A, Vandepoele K, Wasserman WW, Parcy F, Kundaje A, Mathelier A JASPAR 2026: expansion of transcription factor binding profiles and integration of deep learning models Nucleic Acids Res. 2026 Jan 6;54(D1):D184-D193.; doi: 10.1093/nar/gkaf1209

PlantPan

Chow CN, Yang CW, Wu NY, Wang HT, Tseng KC, Chiu YH, Lee TY, Chang WC. 2024. PlantPAN 4.0: updated database for identifying conserved non-coding sequences and exploring dynamic transcriptional regulation in plant promoters. Nucleic Acids Research 52: D1569–D1578. https://doi.org/10.1093/nar/gkad945

PlantGenIE

Sundell D, Mannapperuma C, Netotea S, Delhomme N, Lin YC, Sjödin A, Van de Peer Y, Jansson S, Hvidsten TR, Street NR. 2015. The Plant Genome Integrative Explorer Resource: PlantGenIE.org. New Phytologist 208: 1149–1156. https://doi.org/10.1111/nph.13557

CoGe

Lyons E, Pedersen B, Kane J, Alam M, Ming R, Tang H, Wang X, Bowers J, Paterson A, Lisch D, Freeling M. 2008. Finding and comparing syntenic regions among Arabidopsis and the outgroups papaya, poplar, and grape: CoGe with rosids. Plant Physiology 148: 1772–1781. https://doi.org/10.1104/pp.108.124867

TAIR

Reiser L, Bakker E, Subramaniam S, Chen X, Sawant S, Khosa K, Prithvi T, Berardini TZ. 2024. The Arabidopsis Information Resource in 2024. Genetics 227: iyae027. https://doi.org/10.1093/genetics/iyae027

2. Please also make sure to align figures so that the same order of model system (Birch and hybrid-aspen) occur in the figure and in the legend - this is relevant for figures 1 and 2.

We have updated Figures 1 and 2 to make sure that Populus tremula x tremuloides data are displayed in panel A of both Figure 1 and 2, while Betula pendula data in panel B of the Figures 1 & 2. We made sure that figure legends list the species correspondence to the panels accordingly.

3. In Figure 1 for EIN2 in the hybrid 24h - the authors should carefully review p-value and post hoc letters ... one or the other must be incorrect as there are only A, and ABs and no Bs in the boxplot.

We thank the reviewer for noticing the discrepancy. We have corrected the post-hoc test letters in Figure 1 and Supplemental figure 7.

4. In Figure 2 in which the problem re-occurs with different order of birch and poplar mentioned in the legend and presented in the figure. In addition, the heat maps follow different rankings so neither treatment nor gene occur in the same order in the heatmap clusters. This difference in orders should be emphasised in the figure legend, or alternatively the heatmaps could be reorganised with fixed ranking of treatment and gene.

---

## [Editor Report · Decision Letter 3]

25 Feb 2026

Species specific marker genes for systemic defence and stress responses to leaf wounding and flagellin stimuli in hybrid aspen and silver birch

PONE-D-25-43089R3

Dear Dr. Orlovskis,

We’re pleased to inform you that your manuscript has been judged scientifically suitable for publication and will be formally accepted for publication once it meets all outstanding technical requirements.

Kind regards,

Benedicte Riber Albrectsen

Academic Editor

PLOS One
---

## [Editor Report · Acceptance letter]

PONE-D-25-43089R3

PLOS One

Dear Dr. Orlovskis,

I'm pleased to inform you that your manuscript has been deemed suitable for publication in PLOS One. Congratulations! Your manuscript is now being handed over to our production team.

Kind regards,

on behalf of

Dr. Benedicte Riber Albrectsen

Academic Editor

PLOS One